# A liquid-like organelle at the root of motile ciliopathy

**Ryan L Huizar[1†], Chanjae Lee[1†], Alexander A Boulgakov[1], Amjad Horani[2], Fan Tu[1], Edward M Marcotte[1], Steven L Brody[3], John B Wallingford[1]\***

[1]Department of Molecular Biosciences, University of Texas, Austin, United States; [2]Department of Pediatrics, Washington University School of Medicine, St Louis, United States; [3]Department of Medicine, Washington University School of Medicine, St Louis, United States

**Abstract** Motile ciliopathies are characterized by specific defects in cilia beating that result in chronic airway disease, subfertility, ectopic pregnancy, and hydrocephalus. While many patients harbor mutations in the dynein motors that drive cilia beating, the disease also results from mutations in so-called dynein axonemal assembly factors (DNAAFs) that act in the cytoplasm. The mechanisms of DNAAF action remain poorly defined. Here, we show that DNAAFs concentrate together with axonemal dyneins and chaperones into organelles that form specifically in multiciliated cells, which we term DynAPs, for dynein axonemal particles. These organelles display hallmarks of biomolecular condensates, and remarkably, DynAPs are enriched for the stress granule protein G3bp1, but not for other stress granule proteins or P-body proteins. Finally, we show that both the formation and the liquid-like behaviors of DynAPs are disrupted in a model of motile ciliopathy. These findings provide a unifying cell biological framework for a poorly understood class of human disease genes and add motile ciliopathy to the growing roster of human diseases associated with disrupted biological phase separation.

DOI: https://doi.org/10.7554/eLife.38497.001

**\*For correspondence:**
wallingford@austin.utexas.edu

[†]These authors contributed equally to this work

**Competing interests:** The authors declare that no competing interests exist.

## Introduction

Motile cilia are microtubule-based cellular projections that beat in an oriented manner to generate fluid flows that are critical for development and homeostasis (*Figure 1A*). Accordingly, genetic defects that disrupt motile cilia function are associated with the motile ciliopathy syndrome known as primary ciliary dyskinesia (PCD; MIM 244400) (*Horani et al., 2016*; *Mitchison and Valente, 2017*). PCD is a rare inherited disease that results in repeated sinopulmonary disease, bronchiectasis, cardiac defects such as heterotaxy, situs anomalies, and infertility. Lung disease is the predominant feature of this syndrome, with significant morbidity, and can result in end-stage lung disease requiring lung transplantation (*Horani et al., 2016*). PCD is caused by mutations in at least 40 different genes, and the function of many remains unclear. There is no cure for PCD, so understanding the genetic control of motile cilia assembly and function is an important challenge.

PCD can arise from mutation of any one of the genes encoding subunits of the multi-protein dynein motors that drive ciliary beating (*Figure 1A*, pink) (*Horani et al., 2016*; *Mitchison and Valente, 2017*). These so-called axonemal dyneins are similar in structure and function to cytoplasmic dyneins but are encoded by distinct genes. Interestingly, axonemal dynein motors are pre-assembled in the cytoplasm before deployment to cilia (*Fowkes and Mitchell, 1998*), and it is now clear that PCD also arises from mutations in genes encoding any of an array of cytoplasmic proteins collectively known as dynein axonemal assembly factors (DNAAFs) (*Figure 1A*, blue) (*Desai et al., 2018*).

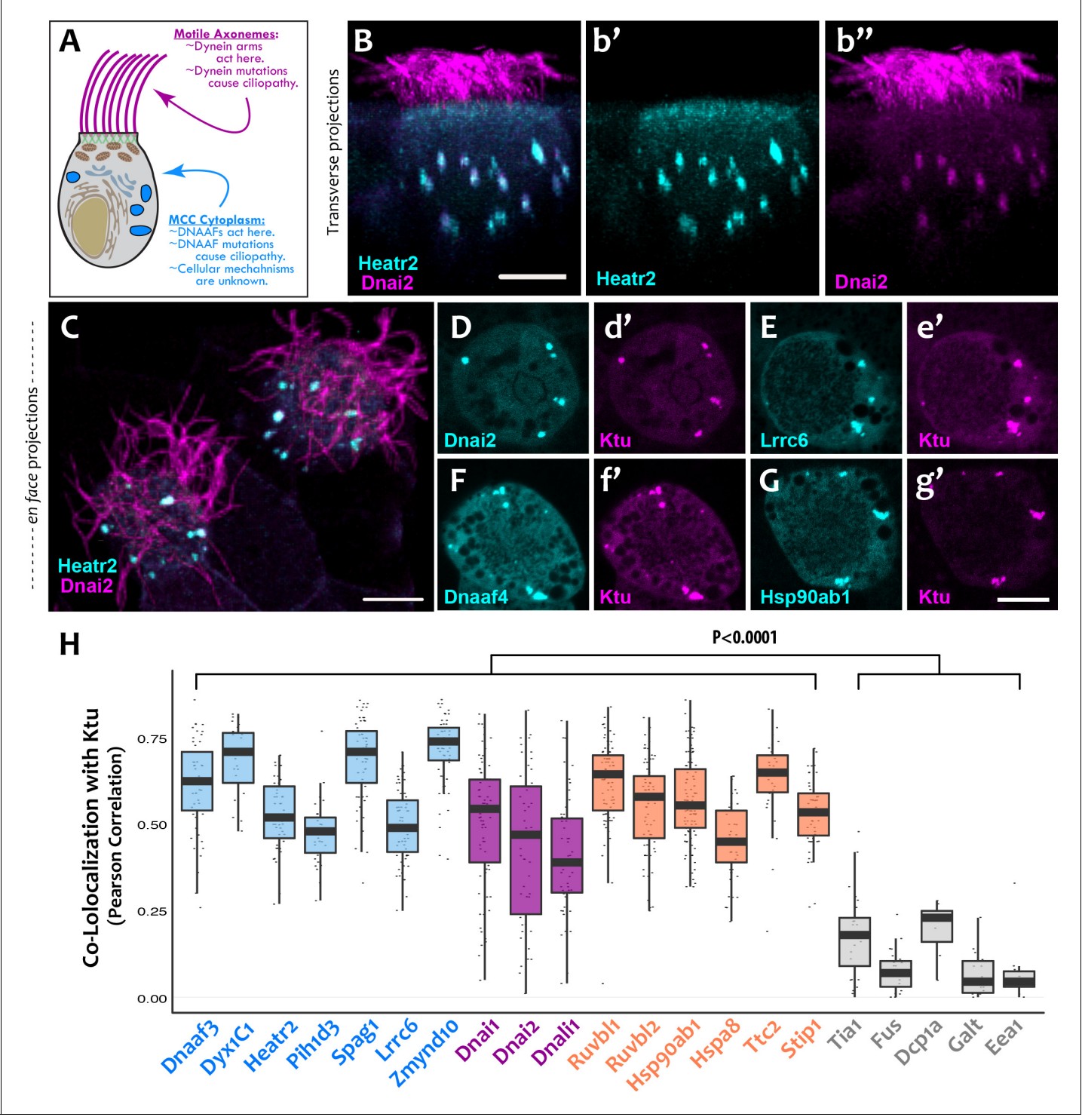

**Figure 1.** DNAAFs, Dyneins, and chaperones co-localize together in DynAPs. (A) Schematic showing a multiciliated cell (MCC) indicating the site of function for proteins linked to motile ciliopathy. Proteins and events in the cytoplasm are indicated in blue; those in the axonemes are indicated in magenta (B) Cross-sectional projection of mucociliary epithelium; GFP-Heatr2 is localized in MCCs to cytoplasmic foci (b'), whereas mCherry-Dnai2 localizes to both axonemes and cytoplasmic foci (b''). (C) *En face* projection showing mCherry-Dnai2 localization in motile axonemes and cytoplasmic foci (magenta), with GFP-Heatr2 showing restricted localization to cytoplasmic foci. (D–G) *En face* optical sections showing co-localization of fluorescent protein (FP) fusions to the indicated proteins. (H) Graph displaying Pearson Correlation Coefficients for colocalization of GFP fusion proteins with mCherry-Ktu at DynAPs. Scale bars 10 μm. p < 0.0001 by one-way ANOVA and post-hoc Tukey-Kramer HSD test. n-values for each FRAP experiment can be found in *Table 2*.

*Figure 1 continued on next page*

*Figure 1 continued*

DOI: https://doi.org/10.7554/eLife.38497.002

The following source data and figure supplements are available for figure 1:

**Source data 1.** Colocalization results presented in *Figure 1H* and *Figure 1—figure supplement 2*.
DOI: https://doi.org/10.7554/eLife.38497.008
**Figure supplement 1.** DNAAFs and Dyneins co-localize in DynAPs in primary human MCC.
DOI: https://doi.org/10.7554/eLife.38497.003
**Figure supplement 2.** Analysis of co-localization in DynAPs.
DOI: https://doi.org/10.7554/eLife.38497.004
**Figure supplement 3.** For unbiased assessment of co-localization, custom software was developed for automated detection of foci in confocal stacks and calculation of 3D co-localization in foci (see Materials and methods).
DOI: https://doi.org/10.7554/eLife.38497.005
**Figure supplement 4.** DynAPs are specific cellular compartments.
DOI: https://doi.org/10.7554/eLife.38497.006
**Figure supplement 5.** DynAPs labeled by endogenous Ruvbl2 are present only in MCCs.
DOI: https://doi.org/10.7554/eLife.38497.007

The initial description of DNAAFs came with the identification of *KTU* (aka *DNAAF2*), which encodes a novel protein present in the cytoplasm yet causes motile ciliopathy when mutated (*Omran et al., 2008*). As in human patients, mutation of *ktu* in fish and the green algae *Chlamydomonas* also elicits defects specifically in cilia beating (*Omran et al., 2008*). Subsequent studies of motile ciliopathy patients defined several additional cytoplasmic DNAAFs, and in all cases, these genes encode cytoplasmic proteins but their mutation results in loss of dyneins specifically from the axonemes, and as a result, defects in cilia beating (*Diggle et al., 2014*; *Horani et al., 2012*; *Horani et al., 2013*; *Kott et al., 2012*; *Mitchison et al., 2012*; *Moore et al., 2013*; *Olcese et al., 2017*; *Paff et al., 2017*; *UK10K et al., 2013*; *Zariwala et al., 2013*). Despite their well-known role in human disease, the molecular mechanisms of DNAAF action are only now emerging.

Proteomic experiments initially indicated a link between DNAAFs and heat shock family chaperones (*Omran et al., 2008*) and many of the DNAAFs contain PIH, TAH1, or CS domains that are common in heat shock co-chaperones (*Olcese et al., 2017*; *Omran et al., 2008*; *Paff et al., 2017*; *UK10K et al., 2013*; *Yamamoto et al., 2010*). Moreover, genetic ablation of the Hsp90 co-chaperone Ruvbl1/2 results in defective dynein arm assembly in zebrafish and mice (*Li et al., 2017*; *Zhao et al., 2013*). More recently, work in diverse model animals has led to a model in which DNAAFs function as part of a 'chaperone relay,' with distinct DNAAFs mediating specific steps in the dynein assembly process (*Cho et al., 2018*; *Mali et al., 2018*; *Olcese et al., 2017*; *Yamaguchi et al., 2018*; *Zur Lage et al., 2018*). Several of these studies suggest that the DNAAFs and chaperones act together in cytosolic 'foci' (e.g. *Diggle et al., 2014*; *Horani et al., 2012*; *Li et al., 2017*), and it has been suggested that different steps in the chaperone relay are somehow compartmentalized (*Mali et al., 2018*), but the nature and mechanisms remain to be defined.

Here, we use live imaging to show that DNAAFs, dynein subunits and chaperones all co-localize to discrete, multiciliated cell (MCC)-specific organelles that display hallmarks of biological phase separation. Remarkably, while DNAAFs and chaperones flux through rapidly, dynein subunits are very stably retained in these organelles, suggesting that these organelles may serve as a specialized compartment for the proposed dynein/chaperone relay system. Finally, in an animal model of motile ciliopathy, altered liquid-like behavior was associated with disrupted assembly and a failure of ciliary beating. Thus, our data identify a novel cell type-specific, liquid-like organelle, provide a cell biological mechanism unifying an emerging class of disease proteins, and suggest a key role for biological phase separation in the etiology of motile ciliopathies.

## Results

### DNAAFs and dynein subunits concentrate in cytosolic foci in multiciliated cells

We found that the DNAAF Heatr2 is present in cytosolic foci in human airway multiciliated cells (MCCs) (*Horani et al., 2012*), and we recently showed that these foci also contain dynein subunits (*Horani et al., 2018*). We have found that another DNAAF, LRRC6 (*Horani et al., 2013*; *Kott et al., 2012*; *Serluca et al., 2009*) is likewise present in dynein-containing cytosolic foci (*Figure 1—figure supplement 1*), but imaging of these structures in human MCCs has proven challenging. We therefore turned to the MCCs of *Xenopus* embryos, which are highly amenable to live imaging, are molecularly tractable, and reliably model the biology of mammalian MCCs (*Walentek and Quigley, 2017*). In *Xenopus* MCCs, GFP-Heatr2 was easily detectable in cytosolic foci in both transverse and *en face* projections of 3D confocal image stacks (*Figure 1B,b'*; C; *Video 1*). These foci were also labeled by mCherry-Dnai2, an outer arm dynein that also strongly labeled axonemes (*Figure 1B,b''*; C). To ask if such foci are a general feature of DNAAFs, we examined the canonical DNAAF Ktu, and found that it too co-localized with Dnai2 in cytosolic foci (*Figure 1D,d'*). As in human MCCs, Lrrc6 displayed a punctate pattern in *Xenopus* MCCs (*Figure 1E*).

To ask if these different proteins were present in a common compartment, we calculated Pearson Correlations for co-localization (*Dunn et al., 2011*), finding strong co-localization in *en face* projections through the MCC cytoplasm for Heatr2, Lrrc6, and Dnai2 as compared to Ktu (*Figure 1H*); the Costes algorithm (*Costes et al., 2004*) demonstrated that these co-localizations were statistically significant (*Figure 1—figure supplement 2*). Because measures of co-localization can be biased by manual selection of regions of interest in the data, we also developed custom software for automated, object-based detection of cytosolic foci and quantification of co-localization in three dimensions (*Figure 1—figure supplement 3A*; see Materials and methods). Using this method, we again observed strong co-localization between the DNAAFs and Dyneins (*Figure 1—figure supplement 3B*). As a negative control for this approach, we computationally randomized the position of foci within these cells, generating ten independent randomized replicates per cell; we found essentially no co-localization after randomization (*Figure 1—figure supplement 3C*).

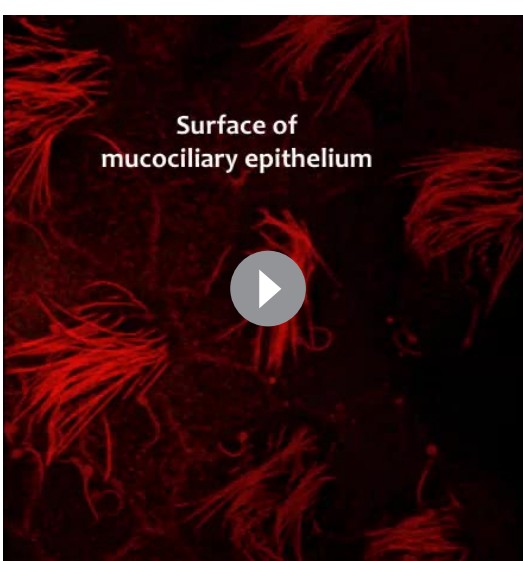

**Video 1.** DynAPs form specifically in the cytoplasm of multiciliated cells. This annotated movie shows a rotating nearest-point projection of a 3D confocal stack of a section of mucociliary epithelium labeled by expression of membrane-RFP (red) and GFP-Ktu (green).
DOI: https://doi.org/10.7554/eLife.38497.009

We next performed a series of control experiments to ask if the cytosolic foci we observed were specific structures, because fluorescent protein (FP) fusions have been found to aggregate non-specifically in some cases (e.g. *Landgraf et al., 2012*). First, we found that DNAAFs did not co-localize with FP-fusions to several known foci-forming proteins, including the P-body marker Dcp1a or the amyloid forming protein Fus (*Kedersha et al., 2005*; *Patel et al., 2015*) (*Figure 1—figure supplement 4A,B,D*). These proteins displayed significantly lower Pearson correlation with Ktu, as compared to DNAAFs or dyneins (*Figure 1H*) and also displayed non-significant Costes values for co-localization with Ktu (*Figure 1—figure supplement 2A*). Second, other components of motile axonemes also require cytoplasmic assembly factors (*Desai et al., 2018*), but FP-fusions to several radial spoke and docking complex-related proteins did not localize in cytosolic foci (*Figure 2A,B*). Finally, a recent screen examining the localization of roughly 200 FP-fusions in *Xenopus* MCCs suggests non-specific aggregation of FPs is rare in these cells (*Tu et al., 2017*).

Our data therefore suggest that the DNAAF- and dynein-containing foci represent novel and

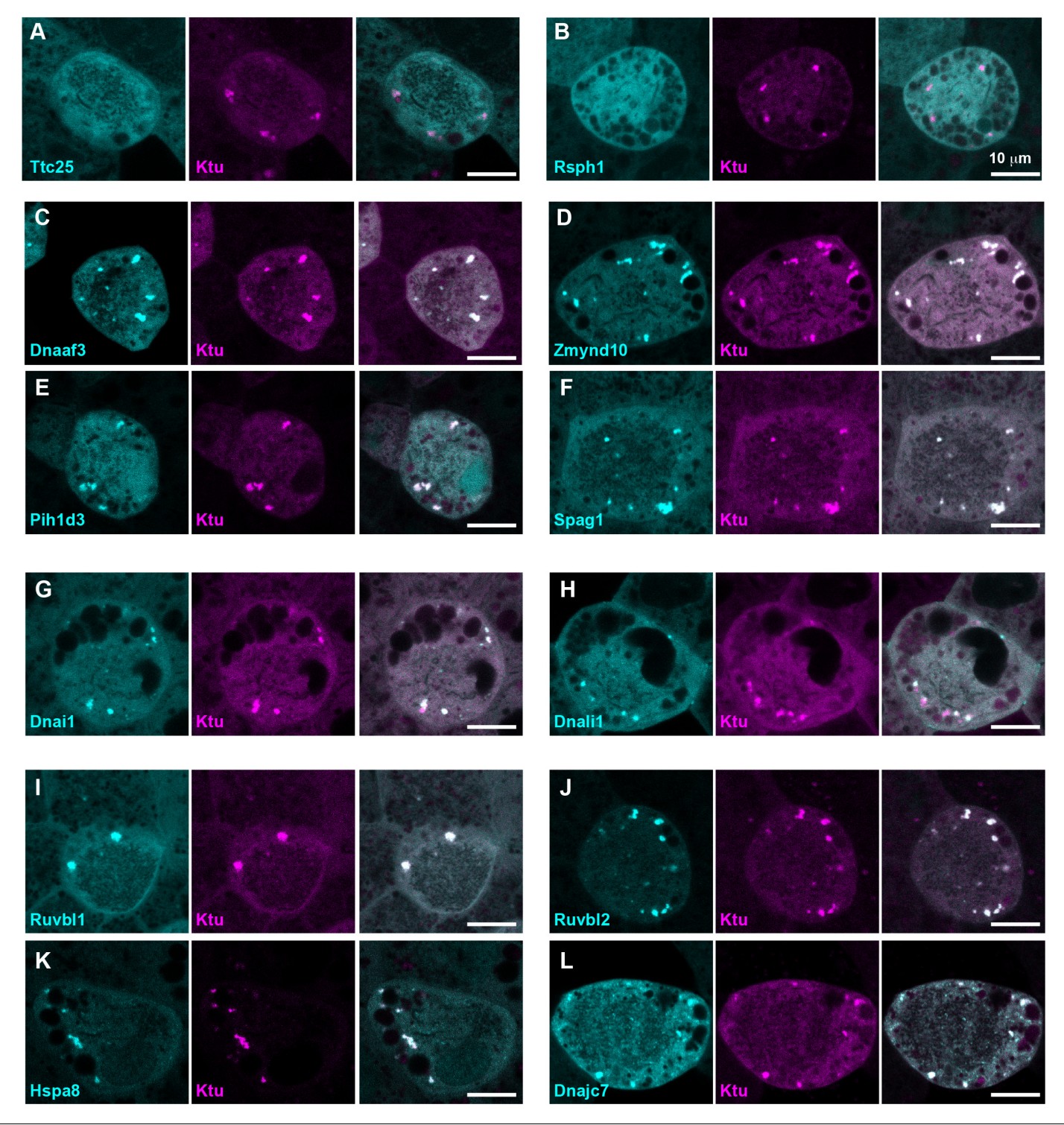

**Figure 2.** DynAPs concentrate DNAAFs, dynein subunits, and Hsp70/90 chaperones. All left panels show GFP fusions (cyan) to indicated proteins, note all co-localize in DynAPs with the mCherry-Ktu (magenta) shown in middle panels. All right panels are merged views. The docking complex protein Ttc25 (panel A) and the radial spoke protein Rsph1 (panel B) serve as negative controls and do not localize in DynAPs. All other do co-localize; see quantifications in *Figure 1H* and *Figure 1—figure supplements 2* and *3*. Scale bars = 10 um.

DOI: https://doi.org/10.7554/eLife.38497.010

specific structures, so we examined FP fusions to seven additional DNAAFs. Consistent with genetic studies demonstrating that all DNAAFs share a common function, we found that all tested DNAAFs co-localized significantly in cytosolic foci (*Figure 2C–F*; *Figure 1H*). In all cases, these foci were irregularly shaped, measured roughly 1–3 microns across, were typically present in the middle third of the apicobasal axis of MCCs, and in *en face* optical sections tended to reside near the cell periphery (*Figure 1B–G*, *Figure 2C–F*). Additional subunits of both inner and outer arm dyneins were also present in these foci (*Figure 2G,H*; *Figure 1H*). Together, these data demonstrate that nine distinct DNAAFs co-localize with axonemal dynein subunits in a common compartment in the MCC

**Table 1.** List of proteins investigated in this study, their functions, and their localization.

| | Protein | Localization | Function |
|---|---|---|---|
| Ciliopathy Associated DNAAFs | Ktu (Dnaaf2) | DynAPs | DNAAF |
| | Dnaaf3 | DynAPs | DNAAF |
| | Dyx1C1(Dnnaf4) | DynAPs | DNAAF |
| | Heatr2 (Dnaaf5) | DynAPs | DNAAF |
| | Lrrc6 | DynAPs | DNAAF |
| | Zmynd10 | DynAPs | DNAAF |
| | Spag1 | DynAPs | DNAAF |
| | Pih1d3 | DynAPs | DNAAF |
| | C21orf59 | DynAPs | DNAAF |
| Dynein Arm Subunits | Dnai1 | DynAPs | Outer Dynein Arm |
| | Dnai2 | DynAPs | Outer Dynein Arm |
| | DnaL1 | DynAPs | Outer Dynein Arm |
| | Dnal4 | DynAPs | Outer Dynein Arm |
| | Dnali1 | DynAPs | Inner Dynein Arm |
| | Wdr78 | DynAPs | Inner Dynein Arm |
| Hsp Family Chaperone-Related | Hsp90ab1 | DynAPs | Hsp90 Chaperone |
| | Hspa8 | DynAPs | Hsp70 Chaperone |
| | Ruvbl1 (Pontin) | DynAPs | Hsp90 Co-chaperone (R2TP) |
| | Ruvbl2 (Reptin) | DynAPs | Hsp90 Co-chaperone (R2TP) |
| | DnajC7 (Ttc2) | DynAPs | Hsp70 Co-chaperone |
| | Stip1 (Hop) | DynAPs | Hsp70 Co-chaperone |
| | Ttc9c | DynAPs | Hsp Co-chaperone |
| Other MCC-Related | Centrin4 | Baseal Bodies | Basal Bodies |
| | Ccdc63 | Other foci | Deuterosome marker |
| | Mns1 | Baseal Bodies | Docking complex-related |
| | Armc4 | Axoneme | Docking complex-related |
| | Ttc25 | Cytosol | Docking complex-related |
| | Rsph1 | Cytosol | Radial Spoke |
| | Nme5 | Cytosol | Radial Spoke |
| Known Foci-Forming | Dcp1a | Other foci | P-body marker |
| | Lsm4 | Other foci | P-body marker |
| | Fus | Nuclear foci | Known foci forming |
| | Tia1(Tiar) | Other foci | Stress granule marker |
| | G3bp1 | DynAPs (+other foci) | Stress granule marker |
| Other Organelles | GalT | Golgi | |
| | EEA1 | Other foci | |

DOI: https://doi.org/10.7554/eLife.38497.011

cytoplasm (*Table 1*). Based on the nomenclature for the genes encoding both the dyneins and their assembly factors, we propose the term DynAP, for <u>Dyn</u>ein <u>A</u>xonemal <u>P</u>articles.

## DynAPs concentrate core Hsp70/90 chaperones and specific co-chaperones

As discussed above, DNAAFs act in concert with heat shock chaperones, and genetic mutants in the co-chaperones Ruvbl1 and Ruvbl2 result in defective dynein assembly and cilia beating. We were satisfied therefore to find that immunostaining for endogenous Ruvbl2 also strongly labeled DynAPs in *Xenopus* MCCs (*Figure 1—figure supplement 5*; *Video 2*), consistent with results from zebrafish and mice (*Li et al., 2017*). This result also further demonstrates that these structures are not artifacts of FP fusions. We note that Ruvbl2 immunostaining also reported much smaller foci that were present in both MCCs and neighboring cells (*Figure 1—figure supplement 5*), consistent with previous reports that this broadly acting co-chaperone can form cellular foci (*Rizzolo et al., 2017*). However, these much smaller foci were readily distinguishable from DynAPs by their size.

In addition, we also observed that DynAPs were strongly labeled by FP fusions to other known Hsp co-chaperones, including Stip1 and Dnajc7 (e.g. (*Moffatt et al., 2008*; *Odunuga et al., 2004*)). Finally, we also identified Ttc9c as a DynAP component, which is of interest because the function of this protein is essentially unknown, but it binds Hsp90 and is implicated in cilia beating (*Gano and Simon, 2010*; *Xu et al., 2015*). These data provide a cell biological context for understanding the interplay of axonemal dyneins, ciliopathy-associated DNAAFs, and the ubiquitous machinery of protein folding and homeostasis.

## DynAPs are MCC-specific organelles that assemble under the control of the motile ciliogenic transcriptional circuitry

We next sought to understand the developmental biology of DynAP formation. *Xenopus* MCCs develop and function in concert with other cell types that lack motile cilia, including mucus-secreting goblet cells, ion pumping cells, and seratonin secreting cells (*Walentek and Quigley, 2017*). We reasoned that if DynAPs were dedicated organelles related to axonemal dyneins, they should be cell type specific. To test this idea directly, we ectopically expressed FP-fusions to DNAAFs throughout the mucociliary epithelium (i.e. in MCC as well as other cell types). Strikingly, FP fusions to DNAAFs reported foci only in MCCs and not in adjacent cells lacking motile cilia (*Figure 3A,C,E*; *Figure 1—figure supplement 4D*). This MCC-specificity was not a general property of FP fusions, as the P-body marker Dcp1a assembled into foci in both MCCs and goblet cells (*Figure 1—figure supplement 4D*). Thus, both immunostaining for Ruvbl2 (above) and expression of FP fusions to DNAAFs demonstrate that DynAPs assemble specifically in MCCs and not in neighboring non-ciliated cells.

For a direct test of the cell type-specificity of DynAPs, we turned our attention to the evolutionarily conserved transcriptional circuitry that directs MCC differentiation in vertebrates ranging from *Xenopus* and zebrafish to mice and humans (*Brody et al., 2000*; *Stubbs et al., 2008*; *Stubbs et al., 2012*). At the top of this hierarchy is the master regulator Mcidas, which is necessary and sufficient for the specification of MCCs (*Stubbs et al., 2012*). As previously reported, ectopic expression of Mcidas significantly increased the percentage of MCCs in the

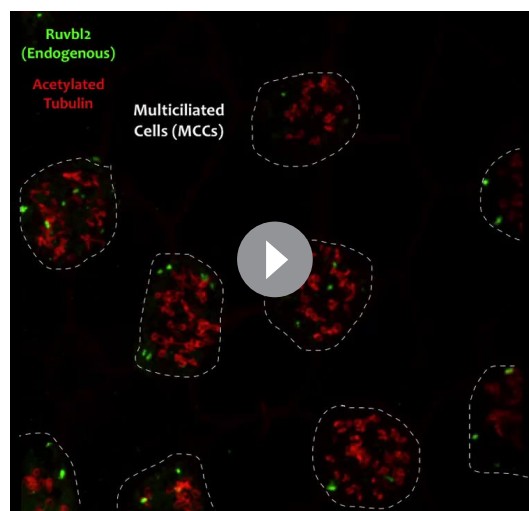

**Video 2.** Endogenous Ruvbl2 is present in DynAPs specifically in the cytoplasm of multiciliated cells. This annotated movie shows a rotating maximum intensity projection of a 3D confocal stack of a section of mucociliary epithelium labeled by immunostaining for acetylated tubulin to label cilia (red) and Ruvbl2 (green). Note that the contrast levels in this projection obscure the smaller Ruvbl2 foci present in both MCCs and gobelt cells (see *Figure 1—figure supplement 5*).
DOI: https://doi.org/10.7554/eLife.38497.012

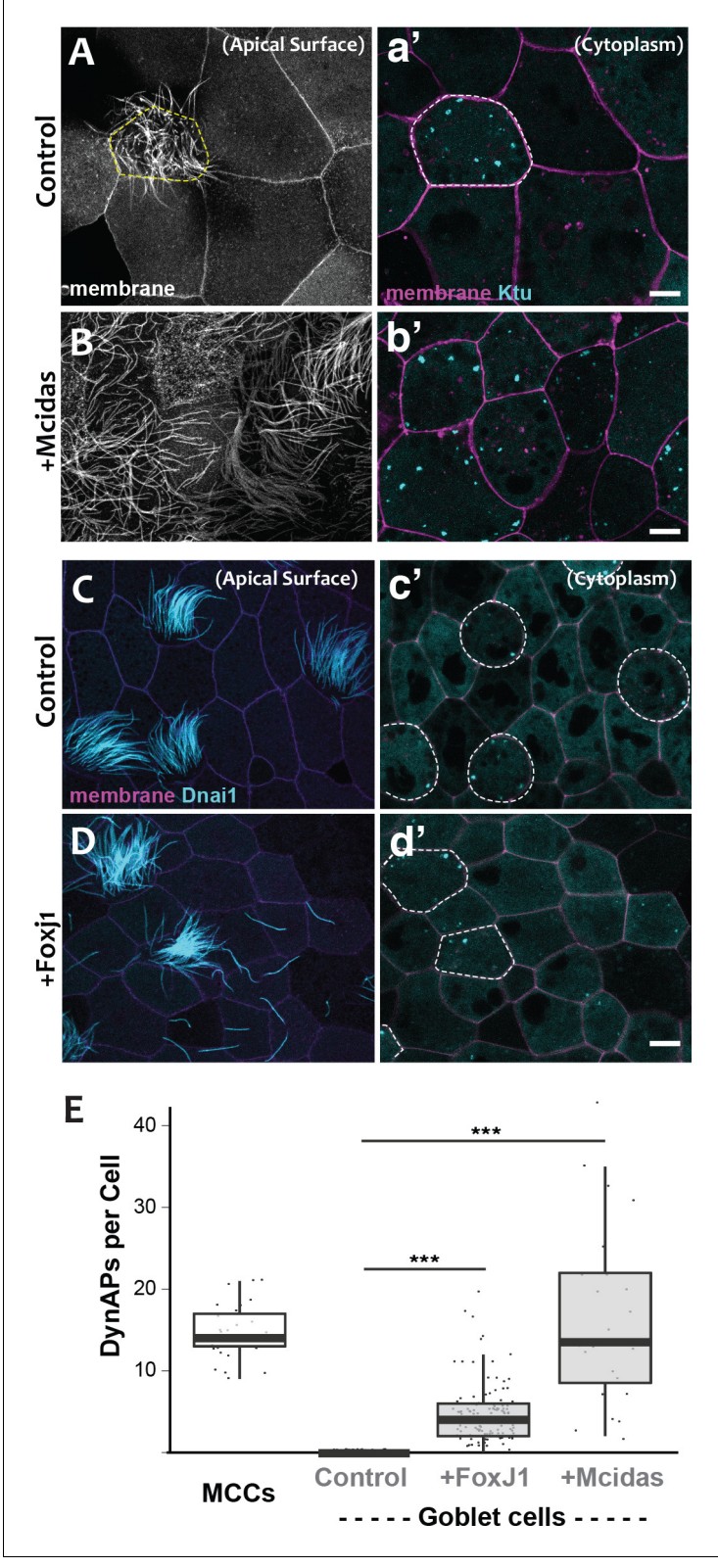

**Figure 3.** DynAPs are MCC-specific and controlled by the motile ciliogenic transcriptional circuitry. (**A**) Membrane labeling with CAAX-RFP at the apical surface reveals a single MCC (cilia, dashed circle) surrounded by non-ciliated goblet cells. (**a'**) Projection of the cytoplasm of the same cells in A. GFP-Ktu is expressed throughout, but forms foci only in the MCC. (**B**) Membrane labeling at the apical surface reveals that expression of Mcidas converts all

*Figure 3 continued on next page*

*Figure 3 continued*

cells to MCCs. (**b'**) Projection of the cytoplasm of the same cells in B; GFP-Ktu forms foci in all cells upon expression of Mcidas.( **C–D**) Apical surface views of a control mucociliary epithelium (**C**) and one ectopic expressing Foxj1 (**D**). (**c'–d'**) Projection of the cytoplasm of the same cells in C and D. GFP-Dnai1 labels both of axonemes at the surface in C, D and DynAPs in cytoplasm in c'd'. CAAX-BFP labels membranes; Dashed lines mark MCCs. Expression of Foxj1 induces solitary ectopic motile cilia (**D**) and ectopic DynAPs (**d'**).(**E**) Graph displaying number of cytoplasmic GFP-Ktu foci in wild-type MCCs and goblet cells as well as ciliogenesis-induced goblet cells. $p < 2.2 \times 10^{-16}$ for Foxj1 and $p = 1.68 \times 10^{-7}$ for Mcidas experiments by two-sample t-test. n = 29 (wild-type MCC), n = 130 (wild-type goblet cell), n = 122 (Foxj1-OE goblet cells), and n = 24 (Mcidas-HGR Goblet cells) across three embryos. Scale bars 10 μm.
DOI: https://doi.org/10.7554/eLife.38497.013
The following source data is available for figure 3:

**Source data 1.** Foci counts for WT MCCs and WT, FoxJ1-OE and Mcidas-OE goblet cells.
DOI: https://doi.org/10.7554/eLife.38497.014

mucociliary epithelium, and importantly, all of these MCCs contained obvious DynAPs (*Figure 3B,b', E*). We then tested the Foxj1 transcription factor, which is essential for motile ciliogenesis and acts downstream of Mcidas (*Brody et al., 2000*; *Stubbs et al., 2008*). Expression of Foxj1 can induce the formation of solitary motile cilia (*Stubbs et al., 2008*), and we found that these ectopic motile cilia were associated with assembly of ectopic DynAPs (*Figure 3C–E*). Thus, DynAPs are specific features of MCCs and their assembly is controlled by the evolutionarily conserved motile ciliogenic transcriptional program.

## DynAPs display hallmarks of biological phase separation

We next explored the cell biological basis of DynAP assembly. DynAPs were not labeled by a general membrane marker (CAAX-RFP) (*Video 1*) or by markers of known membrane-bound organelles such as Golgi or endosomes (*Figure 1—figure supplement 4B,C*). We hypothesized, then, that DynAPs may represent biomolecular condensates formed by liquid-liquid phase separation (*Banani et al., 2017*; *Shin and Brangwynne, 2017*). This possibility was exciting, because while phase separation has emerged as a widespread mechanism for compartmentalizing ubiquitous cellular processes, such as RNA processing and stress responses (*Banani et al., 2017*; *Shin and Brangwynne, 2017*), examples of cytoplasmic phase separated organelles with cell-type specific functions in differentiated somatic cells remain comparatively rare.

One hallmark of phase separated organelles is rapid fission and fusion, such as that observed in *C. elegans* p-granules and mammalian stress granules (*Brangwynne et al., 2009*; *Li et al., 2012*). Time-lapse imaging of DynAPs revealed similar behaviors. DynAPs underwent fission and coalescence on the order of only a few minutes (*Figure 4A*; *Video 3*). A second hallmark of biological phase separation is rapid exchange of material both within organelles and between the organelle and the cytoplasm (*Brangwynne et al., 2009*; *Li et al., 2012*). Using 'half-bleach' FRAP experiments, in which only a portion of the organelle is bleached (*Brangwynne et al., 2009*; *Patel et al., 2015*; *Schmidt and Rohatgi, 2016*; *Woodruff et al., 2017*), we observed rapid intra-DynAP exchange of Ktu (*Figure 4B,C*). GFP-Ktu displayed similarly rapid FRAP kinetics after bleaching of entire DynAPs, suggesting rapid exchange of Ktu between DynAPs and the cytoplasm (*Figure 4B,D*). As a negative control we performed FRAP on the Golgi resident protein GalT, and it displayed no recovery over similar time frames (*Figure 4E*). Thus, both the fission/coalescence behaviors and FRAP kinetics of DynAPs are similar to those observed in organelles known to form via biological phase separation (*Brangwynne et al., 2009*; *Li et al., 2012*; *Patel et al., 2015*; *Schmidt and Rohatgi, 2016*; *Woodruff et al., 2017*).

## DynAPs stably concentrate dynein subunits, while allowing DNAAFs and chaperones to flux through rapidly

Since dyneins are the clients for DNAAFs, we next examined dynein kinetics within DynAPs using FRAP. In stark contrast to Ktu, dynein subunits were stably retained inside DynAPs. We observed very limited mobility within DynAPs in 'half-bleach' FRAP experiments and between DynAPs and the cytoplasm when entire organelles were bleached (*Figure 5A,B,b'*). Similar results were obtained

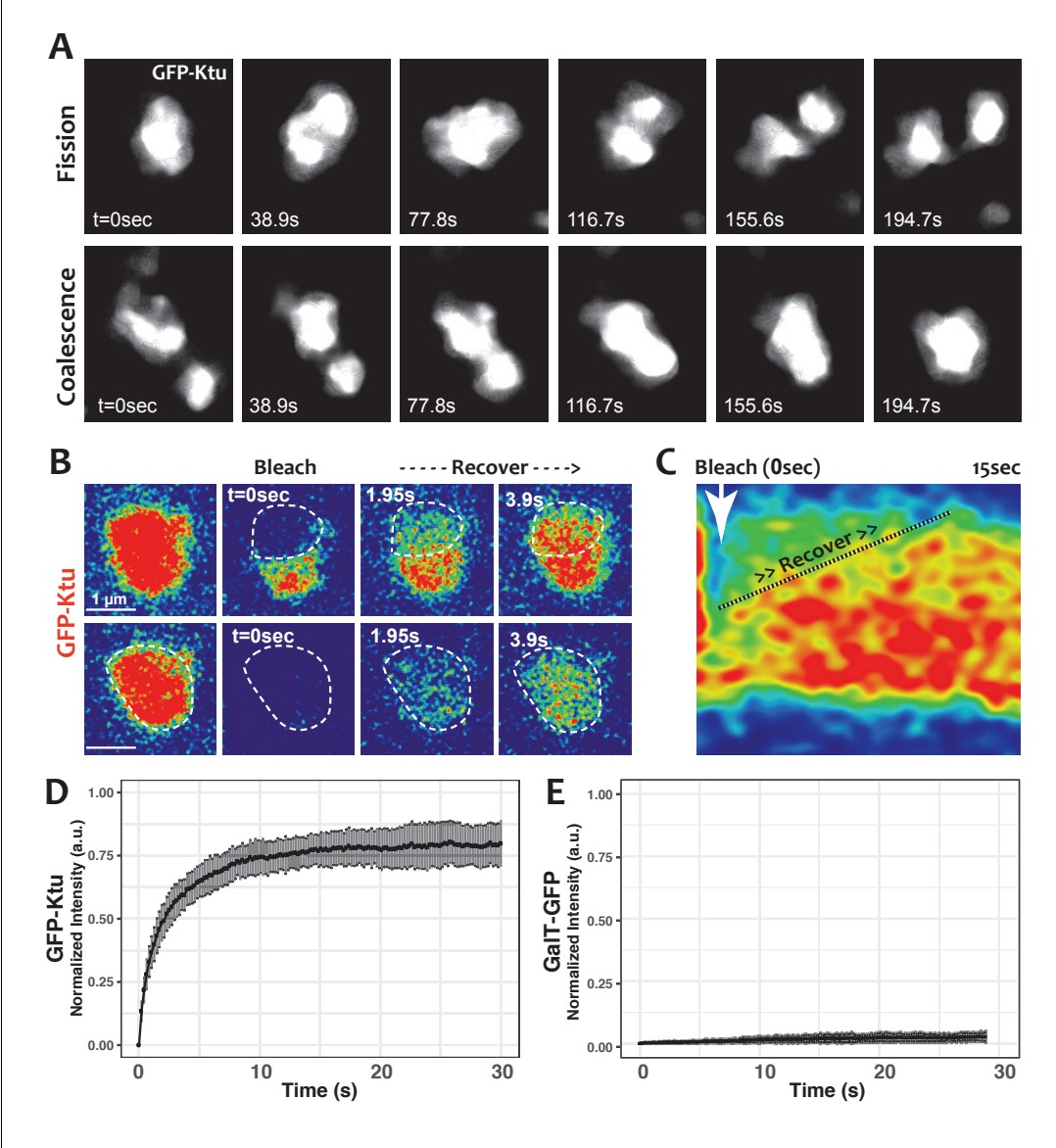

**Figure 4.** DynAPs display liquid-like behaviors. (**A**) Stills from time-lapse imaging of an individual DynAP labeled with GFP-Ktu undergoing fission (upper) and later coalescence (lower) (time in seconds). (**B**) Time-lapse images of GFP-Ktu recovery after photobleaching of partial (upper) or entire DynAPs (lower). Images are color-coded to highlight changes in pixel intensity (blue = low; red = high; green = intermediate. Dashed line marks the photobleaching area. (**C**) Kymograph of GFP-Ktu recovery after partial bleach, showing that recovery of the bleached area occurs from within the DynAP, rather than from the cytoplasm. Note, however, that FRAP of whole DynAPs indicates that rapid exchange also occurs between DynAPs and the cytoplasm (Panel B, lower; and see *Table 2*). (**D**) FRAP kinetics of GFP-Ktu after bleaching entire DynAPs. (**E**) FRAP kinetics of the *trans*-golgi protein GalT-GFP after bleaching golgi-derived vesicles.

DOI: https://doi.org/10.7554/eLife.38497.015

The following source data is available for figure 4:

**Source data 1.** Aggregate data for FRAP curves presented in *Figures 4* and *5*.
DOI: https://doi.org/10.7554/eLife.38497.016

with both inner and outer arm dynein subunits (*Figure 5D*). These results prompted us to test other DynAP-localized proteins, and we found that all tested DNAAFs and chaperones displayed significantly higher mobile fractions when compared to all dynein subunits (*Figure 5C,D*; *Table 2*). Thus, DynAPs stably concentrate dynein subunits, while assembly factors and chaperones rapidly flux through.

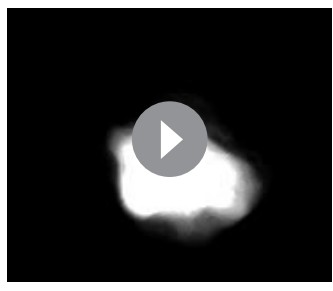

**Video 3.** Liquid-like fission and fusion of a single DynAP. This non-annotated movie shows smoothened data from a time-lapse movie of a single Heatr2-labeled DynAP. Stills from this movie are presented in *Figure 4A*.
DOI: https://doi.org/10.7554/eLife.38497.017

## DynAPs are distinct from stress granules but share a subset of molecular machinery

The mixture of more-fluid and more-stable components we observed in DynAPs is consistent with data from other phase-separated organelles, and recent studies suggest that these organelles frequently share protein machinery (*Aizer et al., 2008*; *Feric et al., 2016*; *Jain et al., 2016*; *Kedersha et al., 2005*). Because stress granules also contain Ruvbl2 and the Hsp70/90 chaperones (*Jain et al., 2016*), we next probed the relationship between DynAPs and stress granules. We found that the canonical stress granule marker G3bp1 (*Kedersha et al., 2005*; *Wheeler et al., 2016*) was strongly enriched in DynAPs, where it co-localized significantly with Ktu (ovals in *Figure 6A,a2*; *Figure 6B*). Curiously, however, G3bp1-FP also consistently labeled a second population of smaller foci in MCCs that did not contain DNAAFs (boxes in *Figure 6A,a2*). FRAP experiments revealed turnover kinetics in both populations of foci that were similar to those reported for G3bp1 in stress granules (*Figure 6C*) (*Kedersha et al., 2005*). Thus, G3bp1 is a component of at least two populations of cytoplasmic foci in MCCs, including DynAPs.

To ask if DynAP localization was a common feature of stress granule proteins, we examined Tia1, which also functions in stress granules (*Kedersha et al., 1999*). Tia1-FP localized to small foci in MCCs similar to the DNAAF-negative foci labeled by G3bp1 (*Figure 1—figure supplement 4E*). However, unlike G3bp1, Tia1-FP was typically not present in DynAPs, though in some cases partial co-localization between Tia1 and DNAAFs was observed (e.g. *Figure 1—figure supplement 4E*,

**Table 2.** FRAP data for all tested proteins.

|  | Protein | Mobile fraction (%) | N |
|---|---|---|---|
| Total Bleach | Ktu | 77.38 ± 6.63 | 25 |
|  | Dnaaf3 | 77.57 ± 6.64 | 6 |
|  | Dnaaf4 | 61.64 ± 10.01 | 14 |
|  | heatr2 | 60.71 ± 6.02 | 14 |
|  | Lrrc6 | 58.53 ± 10.75 | 32 |
|  | Pih1d3 | 61.66 ± 7.14 | 7 |
|  | Ruvbl2 | 61.82 ± 7.14 | 13 |
|  | Hsp90ab1 | 54.60 ± 2.89 | 4 |
|  | Hspa8 | 74.00 ± 9.67 | 16 |
|  | Dnai1 | 25.15 ± 5.76 | 13 |
|  | Dnai2 | 16.88 ± 5.75 | 13 |
|  | Dnali1 | 22.02 ± 5.71 | 5 |
|  | G3bp1 (DynAP) | 84.11 ± 6.43 | 11 |
|  | G3bp1(SG) | 83.93 ± 11.08 | 11 |
| Partial Bleach | Ktu | 66.66 ± 8.43 | 13 |
|  | Ruvbl2 | 55.20 ± 9.59 | 4 |
|  | Dnai2 | 20.10 ± 9.77 | 7 |
|  | G3bp1 (DynAP) | 74.83 ± 12.26 | 12 |
|  | G3bp1(SG) | 88.95 ± 10.06 | 6 |

DOI: https://doi.org/10.7554/eLife.38497.018

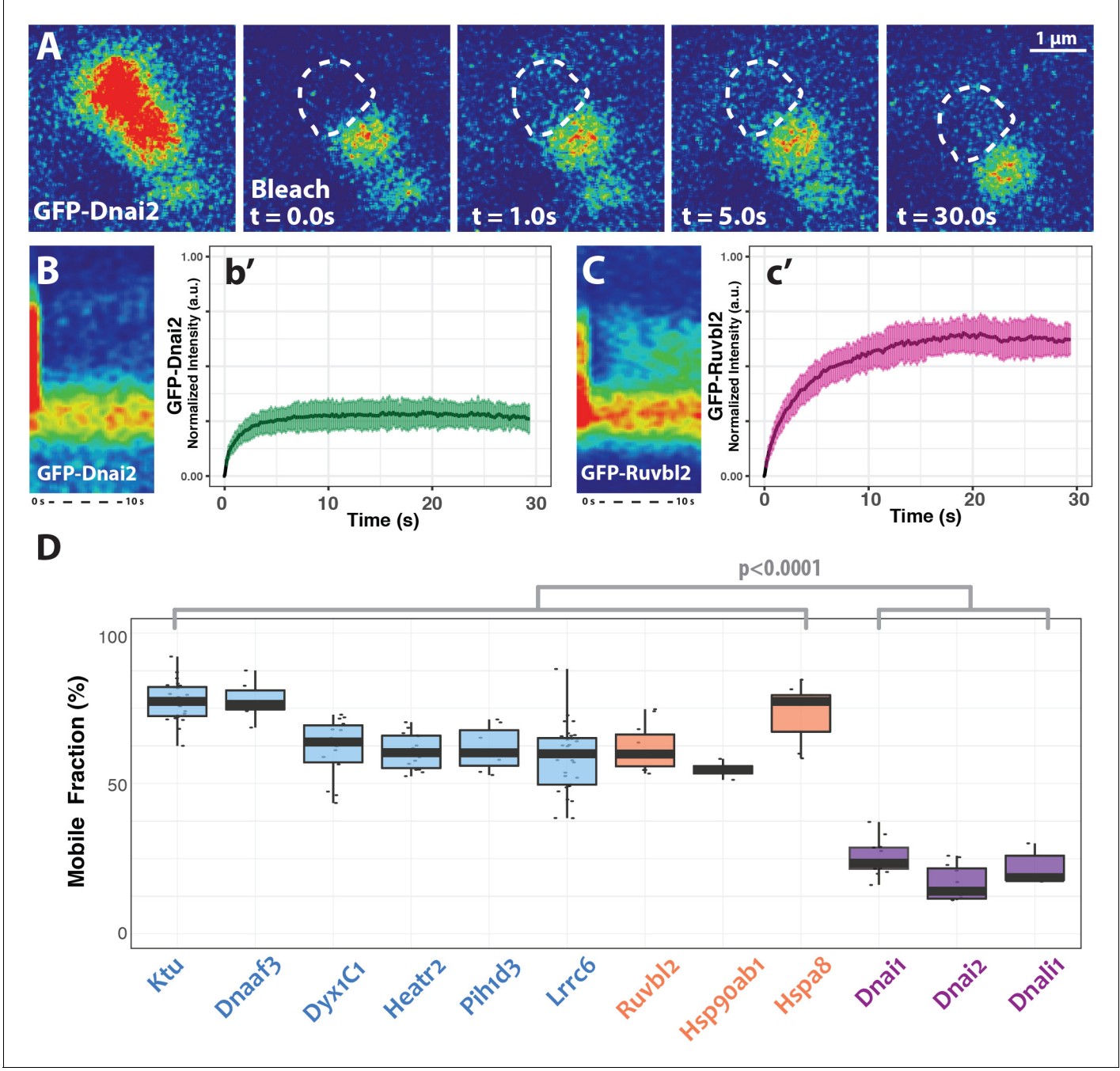

**Figure 5.** DynAP stably retain axonemal dynein subunits. (**A**) Time-lapse images of GFP-Dnai2 recovery after partial photobleaching of a DynAP reveals little recovery after 30 s. Dashed line marks the photobleaching area. (**B**) Kymograph of the first 10 s following photobleaching displays little recovery of GFP-Dnai2. This is reflected by the FRAP kinetics of Dnai2 after complete bleaching of DynAPs (**b'**). (**C**) In stark contrast, the kymograph of the first 10 s following partial photobleaching of the DNAAF GFP-Ruvbl2 displays rapid recovery at DynAPs. This is reflected by the FRAP kinetics of GFP-Ruvbl2 after complete bleaching of DynAPs (**c'**). (**D**) Boxplots of the mobile fractions of various resident proteins at DynAPs. Ciliopathy-related DNAAFs (blue) and canonical chaperones (orange) display greater fluorescence recovery than dynein arm components (purple). p<0.0001 by one-way ANOVA and post-hoc Tukey-Kramer HSD test. n-values for each FRAP experiment can be found in *Table 2*.

DOI: https://doi.org/10.7554/eLife.38497.019

The following source data is available for figure 5:

**Source data 1.** Aggregate data for FRAP curves presented in *Figures 4* and *5*.
DOI: https://doi.org/10.7554/eLife.38497.020

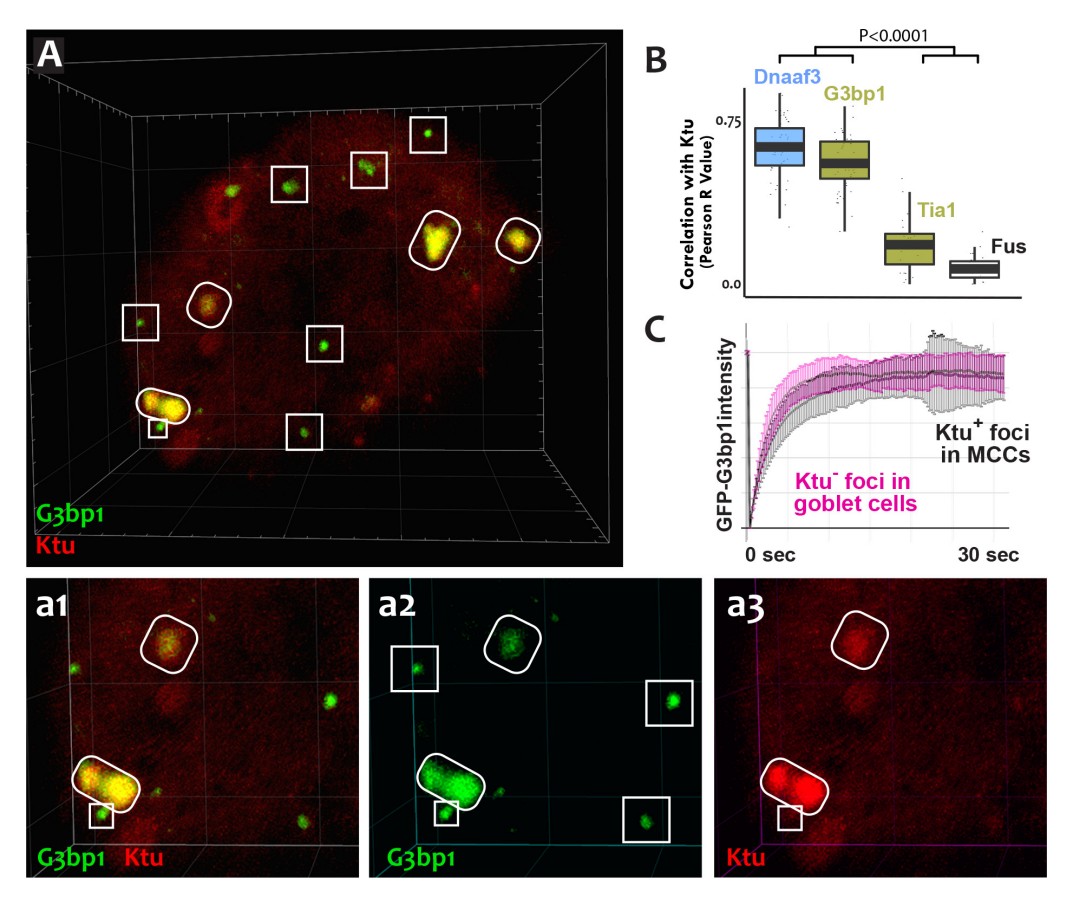

**Figure 6.** DynAPs share molecular and physical properties with stress granules. (**A**) GFP-G3bp1 strongly co-localizes with DNAAFs in DynAPs (ovals), but also labels smaller foci that do not contain DNAAFs (boxes). (**a1–a3**) Higher magnification views of the bottom left corner of the MCC shown in panel A. (**B**) Quantification of co-localization relative to mCherry-Ktu (Dnaaf4 and Fus data from *Figure 1* are recapitulated here for comparison). (**C**) FRAP kinetics of GFP-G3bp1 in Ktu-positive DynAPs (black) in MCCs and in Ktu-negative foci in neighboring goblet cells (pink). p < 0.0001 by one-way ANOVA and post-hoc Tukey-Kramer HSD test. n-values for each FRAP experiment can be found in *Table 2*.
DOI: https://doi.org/10.7554/eLife.38497.021

The following source data is available for figure 6:

**Source data 1.** Data for GFP-G3bp1 FRAP experiments presented in *Figure 6C*.
DOI: https://doi.org/10.7554/eLife.38497.022

arrow). However, the Pearson correlation between Tia1 and Ktu was significantly lower than that observed for G3bp1 or for other DynAP proteins (*Figure 6B*). Thus, despite sharing a similar semi-liquid-like behavior, DynAPs share only a subset of molecular components with stress granules, including G3bp1, Ruvbl1/2, and the Hsp70/90 chaperones (*Jain et al., 2016*).

## Loss of the DNAAF Heatr2 disrupts assembly of DynAPs and alters their liquid-like behavior

Our data suggest that DynAPs share properties with stress granules, but even in those well-studied organelles, it remains unclear just how their liquid like properties impact organelle function. We considered two models for DynAP function. First, given that genetic disruption of either DNAAFs or DynAP-localized chaperones results in specific failure of axonemal dynein assembly, we considered that the kinetics observed in DynAPs could reflect a modified 'reaction crucible' function (*Shin and Brangwynne, 2017*), whereby dynein clients are stably retained as they are acted upon by a procession of assembly factors fluxing through. Consistent with this idea, we found that DynAPs were present not only in mature MCCs but also in nascent MCCs undergoing ciliogenesis (*Figure 7*). On the

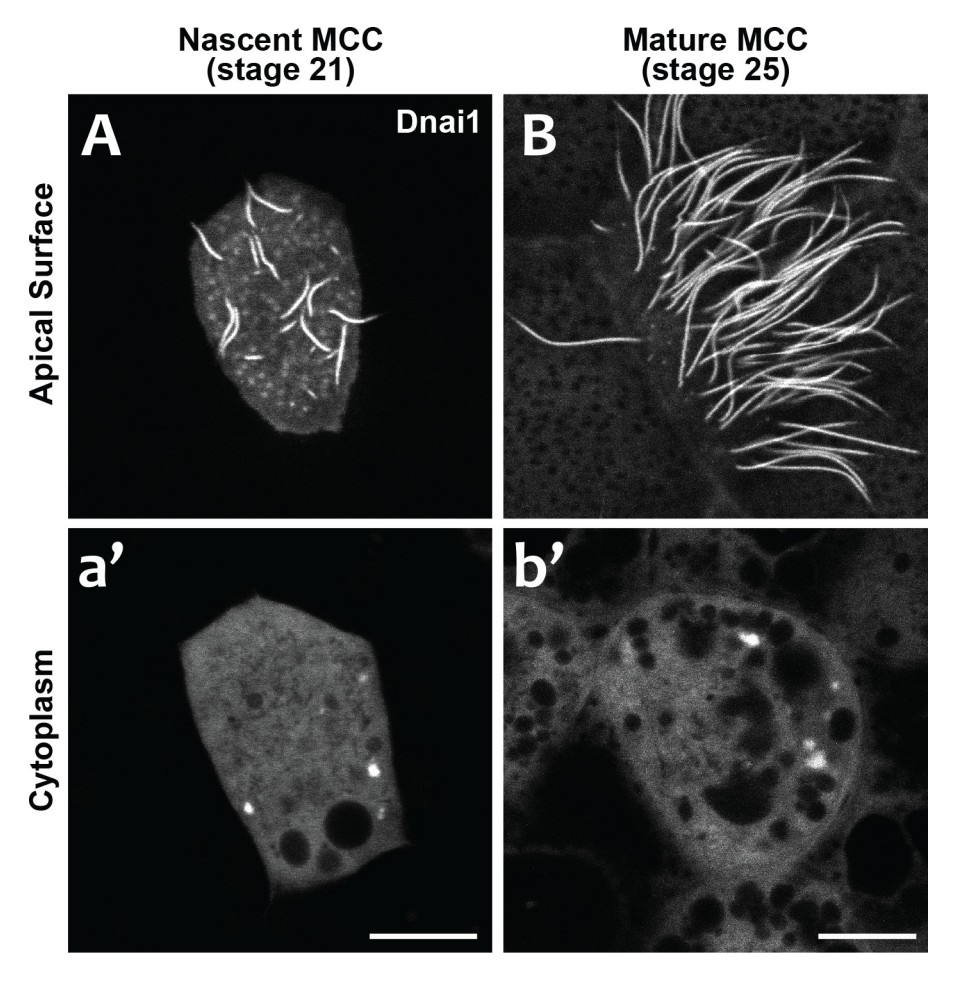

**Figure 7.** DynAPs are present in nascent MCCs during ciliogenesis. (A) GFP-Dnai1 labels the short, still-growing axonemes during MCC ciliogenesis at stage 21. (B) GFP-Dnai1 also labels the longer, mature axonemes at stage 25. (a', b') GFP-Dnai1 labeled DynAPs are present in the cytoplasm at both stages. Scale bars = 10 um.
DOI: https://doi.org/10.7554/eLife.38497.023

other hand, the continued presence of DynAPs in mature MCCs may instead indicate a sequestration function, for example preventing deployment of mis-assembled dynein arms to axonemes during homeostasis.

We reasoned that modeling motile ciliopathy might shed light on DynAP function: If DynAPs are sequestering organelles for mis-assembled dyneins, we expect their numbers to increase if DNAAF function is disrupted; if they are assembly organelles, we expect the converse. To test this idea, we focused on Heatr2, which acts early in the assembly process (*Horani et al., 2018*). Nonsense mutations in *HEATR2* that lead to protein loss cause disease (*Diggle et al., 2014*; *Horani et al., 2012*), and thus can be modeled simply by knockdown in *Xenopus*. To this end, we performed knockdown using morpholino antisense oligonucleotides (MOs). Like all knockdown reagents, MOs have their caveats, so it is important that the MO experiments described below meet the current standards for use (*Blum et al., 2015*; *Eisen and Smith, 2008*; *Stainier et al., 2017*), as MOs targeting two distinct regions of *heatr2* eliminated the mRNA for the targeted gene and the two elicited identical phenotypes that closely recapitulate those observed in humans and flies with genetic mutations.

We designed separate MOs to disrupt splicing of the L and S alloalleles of *heatr2* in the allotetraploid genome of *X. laevis* (*Session et al., 2016*) (see Materials and methods), and RT-PCR demonstrated that co-injection of the two MOs was effective. Not only was splicing severely disrupted, but the overall level of *heatr2* mRNA was also severely reduced, a common effect of nonsense mediated

decay when splicing is disrupted by MOs (*Figure 8—figure supplement 1A*). Time-lapse imaging of cilia beating revealed that the loss of *heatr2* mRNA was accompanied by a severe defect in cilia beating (*Figure 8A,B*). Moreover, confocal imaging of the apical surface of MCCs revealed that while cilia length was not disrupted, Heatr2 knockdown eliminated the axonemal localization of the outer arm dynein Dnai2, (*Figure 8C,D*). A second MO targeting a different splice site in *heatr2* elicited the same phenotypes in *Xenopus*, indicating that this phenotype was not the result of off-target effects (*Figure 8—figure supplement 1*). Thus, MO knockdown targeting either of two distinct sites closely recapitulated the phenotype observed after loss of Heatr2 by genetic mutation in both human PCD patients and in *Drosophila* (*Diggle et al., 2014*; *Horani et al., 2012*).

Strikingly, knockdown with either of the MO sets also elicited a significant reduction in the number of DynAPs (*Figure 8E–G*; *Figure 8—figure supplement 1E–G*). This result suggested a role for Heatr2 in DynAP assembly, which was intriguing because Heatr2 is not implicated in chaperone function, but does act early in the dynein assembly process in human MCCs (*Horani et al., 2018*). Interestingly, changes in protein FRAP mobility accompany pathological alterations of phase separated organelles in neurodegenerative diseases (*Patel et al., 2015*; *Schmidt and Rohatgi, 2016*), so we used FRAP to ask if Heatr2 loss may impact the liquid-like behavior of DynAPs. Indeed, loss of Heatr2 significantly decreased the FRAP mobile fraction of Ktu in DynAPs (*Figure 8H,h'*), suggesting that an alteration in liquid like behavior is linked to defects in DynAP assembly that in turn associate with defects in cilia beating in an animal model of motile ciliopathy.

## Discussion

Here, we have shown that an entire class of motile ciliopathy genes (the DNAAFs) encode proteins that co-localize together with axonemal dynein subunits and chaperones in discrete organelles we term DynAPs (*Table 1*). DynAPs are MCC-specific, form under the control of the conserved genetic circuitry that governs motile ciliogenesis, and display hallmarks of biological phase separation. Moreover, loss of dynein arms from axonemes after disruption of DNAAF function was associated with defective DynAP assembly and altered liquid like character in these organelles. These findings provide a unifying cell biological framework for a poorly understood class of human disease genes and add motile ciliopathy to the growing roster of human diseases associated with disrupted biological phase separation.

As is the case for most phase separated organelles, the significance of concentrating DNAAFs and dyneins into organelles remains a key unanswered question. Nonetheless, our data suggest a model in which client dyneins are concentrated into DynAPs so they can be acted upon by chaperones and co-chaperones that rapidly flux through. This model is parsimonious for integrating three essential functions required for dyneins assembly: First, dyneins are particularly large multi-protein complexes; their assembly takes time and requires multiple assembly factors. Sequestering the client in one compartment would allow the many requisite assembly factors to work together efficiently. Second, large protein complexes require strict quality control, and concentration in DynAPs provides an efficient means to couple assembly and quality control. Third, assembled dyneins might then be stored in DynAPs until they are needed for rapid deployment, which is important because motile cilia are known to very rapidly regenerate.

Finally, our description of DynAPs suggests an attractive hypothesis regarding compartmentalization of the myriad biochemical processes that arise as cell types proliferate in developing embryos. Despite the emergence of phase separation as a mechanism for compartmentalizing cellular functions, cell type-specific, phase separated organelles remain relatively rare, for example Cajal bodies control genome organization in the nuclei of some cell types and the Balbiani body facilitates oocytes' long-term dormancy (*Banani et al., 2017*; *Shin and Brangwynne, 2017*). Our description of DynAP formation specifically in MCCs predicts that a wide range of cell-type specific-liquid like organelles may await discovery. Moreover, our data suggest a model whereby the molecular framework of known ubiquitous liquid-like organelles such as stress granules (e.g. G3bp1) is differentially modified by cell type-specific transcriptional circuits in order to assemble novel organelles to achieve specific functions.

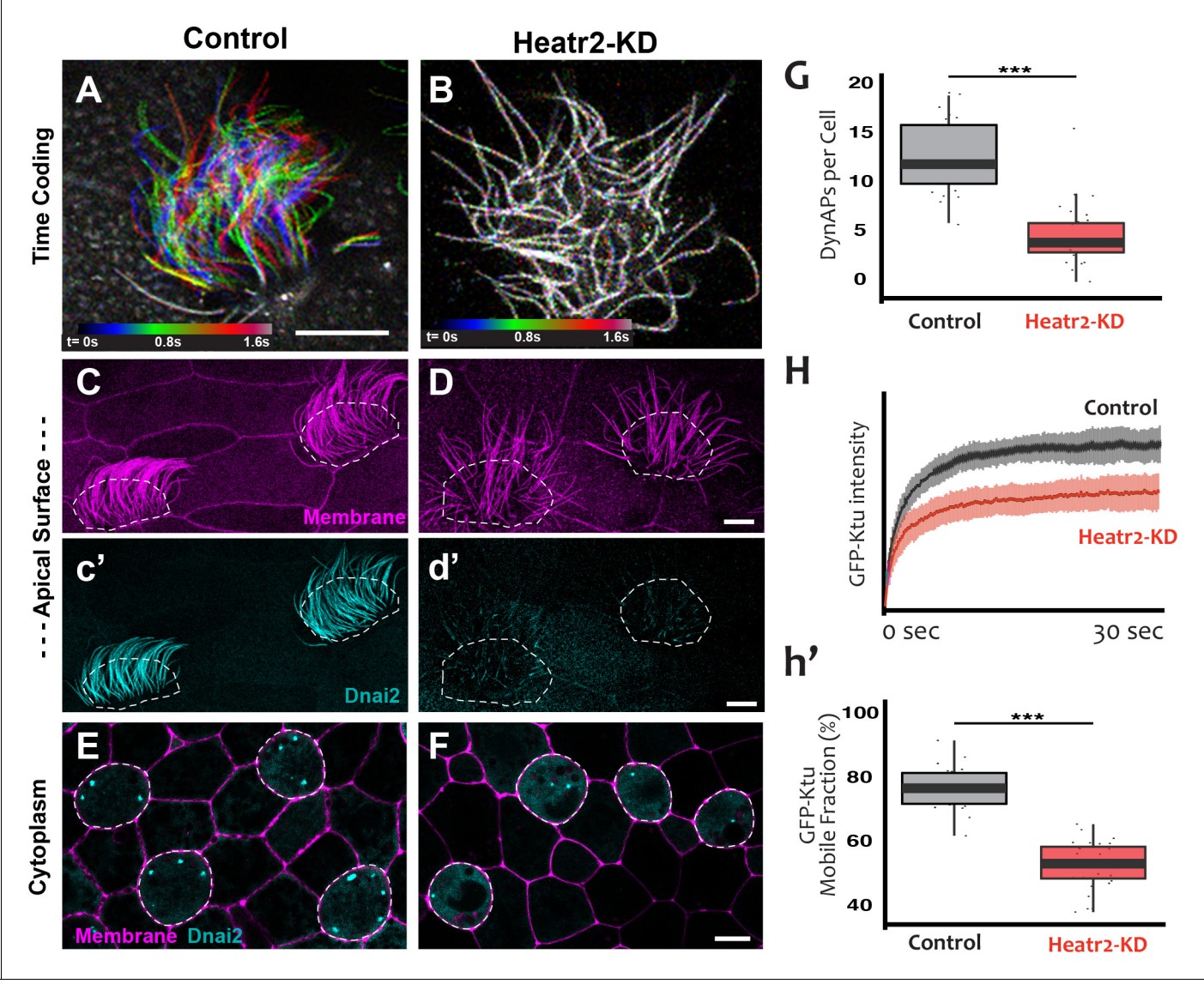

**Figure 8.** Loss of Heatr2 disrupts formation of DynAPs and alters the liquid-like behavior of Ktu. (**A**) Color-based time coding of a high-speed time-lapse movie of an MCC; Successive frames of the movie are color coded as indicated in the time key and overlaid; distinct colors in the overlay reveals ciliary movement. (**B**) Similar time-coding of an MCC after Heatr2-KD; the lack of color reflects the absence of ciliary movement between frames in the movie. (**C**) Membrane labeling with CAAX-RFP (pink) reveals normal cilia morphology in control MCCs (an projection of the confocal optical slices specifically through the apical surface is shown with; dashed lines indicate MCCs. (**c'**) Labelling with GFP-Dnai2 reveals normal localization to the motile axonemes shown in panel C. (**D**) Membrane labeling reveals normal morphology of motile cilia in MCCs after Heatr2 KD. (**d'**) GFP-Dnai2 is lost from the motile cilia shown in panel D. (**E**) Dnai2 is present in DynAPs visible in an *en face* projection *through the cytoplasm* of control MCCs (indicated by dashed lines). (**F**) Despite loss from motile axonemes, Dnai2 remains localized to DynAps in the cytoplasm of MCCs after Heatr2 knockdown. (**G**) Despite the presence of DynAPs in Heatr2-KD MCCs, the number of foci in these cells is significantly reduced relative to wild-type MCCs. $p = 1.27 \times 10^{-10}$ by two-sample t-test (n = 25 wild-type cells, 26 Heatr2-KD cells across two experiments, three embryos each). (**H, h'**) FRAP reveals that Heatr2 knockdown significantly impairs the mobility of GFP-Ktu in DynAPs. Control vs. Heatr2-KD GFP-Ktu mobile fraction, $p < 2.2 \times 10^{-16}$ by two-sample t-test (n = 25 vs. 31 observations, each in independent cells across three experiments, three embryos each). Scale bars 10 µm.

DOI: https://doi.org/10.7554/eLife.38497.024

The following source data and figure supplement are available for figure 8:

**Source data 1.** GFP-Ktu foci counts for WT and Heatr2-MO (Set #1) MCCs.
DOI: https://doi.org/10.7554/eLife.38497.026

**Source data 2.** Kinetic data for GFP-Ktu FRAP experiments in WT and Heatr2-MO MCCs.
DOI: https://doi.org/10.7554/eLife.38497.027

**Source data 3.** GFP-Ktu foci counts for WT and Heatr2-MO (Set #2) MCCs.

*Figure 8 continued on next page*

*Figure 8 continued*

DOI: https://doi.org/10.7554/eLife.38497.028

**Figure supplement 1.** MOs targeting a distinct splice sites in *heatr2* elicited defects in dynein delivery to axonemes and DynAP assembly.

DOI: https://doi.org/10.7554/eLife.38497.025

## Materials and methods

### *Xenopus* embryo manipulations

Female adult *Xenopus* were induced to ovulate by injection of hCG (human chorionic gonadotropin). In vitro fertilization was carried out by homogenizing a small fraction of a testis in 1X Marc's Modified Ringer's (MMR). Embryos were dejellied in 1/3x MMR with 2.5%(w/v) cysteine at pH7.8, microinjected with mRNA or morpholinos (MOs) in 2% ficoll (w/v) in 1/3x MMR. Injected embryos were washed with 1/3x MMR after 2 hr and were reared in 1/3x MMR until the appropriate stages.

### Plasmids, MOs and microinjections

*Xenopus* gene sequences were provided from Xenbase (www.xenbase.org) (*James-Zorn et al., 2018*; *Karimi et al., 2018*) and open reading frames (ORF) of genes were amplified from the *Xenopus* cDNA library by polymerase chain reaction (PCR). The PCR products were inserted into a pCS vector containing a fluorescence tag. The cloned genes are as follows: KTU, Heatr2, DNAI2, DNALI1, ZMYND10, LRRC6, PHI1D1, PHI1D3, Hsp90ab1, TTC9C, STIP1, DnajC7, G3BP1, Fus, and Tia1 into pCS10R-N-term GFP; DNAAF4 and Ruvbl2 into pCS10R-C-term GFP; Ruvbl2, KTU, Heatr2, DNAI2 into pCS10R-N-term mCherry; DCP1a into pCS-dest-mCherry. Ccdc39 and Lsm4 were obtained from the Human ORFeome and DNAAF3, Dyx1c1, and EEA1 were amplified from the *Xenopus* cDNA library were cloned into pCS2 +Gateway destination vectors containing an alpha tubulin promoter and an RFP tag or a GFP tag respectively, via the Gateway LR Clonase II Enzyme. Human GalT-GFP or RFP was derived from GalT-CFP (*Nichols et al., 2001*) by exchange of CFP. Capped mRNAs were synthesized using mMESSAGE mMACHINE SP6 transcription kit (ThermoFisher Scientific, AM1340). Morpholino antisense oligonucleotides (MOs) against *Heatr2* were designed to block splicing of mRNAs transcribed from both L and S alloalleles (Gene Tools). Heatr2 MO sequences and injected doses are as follow:

> Heatr2 MO set #1: Heatr2.L (30 ng): 5'-ACATTATCAATCACAACCTGGTATA-3'
> Heatr2.S (30 ng): 5'-CATTGAATTCCTCACCTGATTTCAG-3'
> Heatr2 MO set #2: Heatr2.L (5 ng): 5'- GGATCATGTAAGACACCTACCTGCA-3'
> Heatr2.S (5 ng): 5'-GGTAAAAAACACCTACCTGAACTGA-3'

For imaging, mRNAs and DNAs for fluorescence proteins were injected into two ventral blastomeres of 4 cell stage embryos with 100 pg/injection and 40 pg/injection, respectively. mRNA of FoxJ1 (*Pohl and Knöchel, 2004*) was injected with 200 pg/injection into two ventral blastomeres. mRNA of MCIDAS(*Stubbs et al., 2012*)-hGR from CS10R-MCIDAS-hGR (100 pg/injection) was injected into all four blastomeres at the 4 cell stage and animal cap explants were dissected at stage 8, treated with $10^{-7}$ M dexamethasone at stage 11, cultured until stage 26 and then imaged. For all experiments, embryos were selected at random from multiple clutches, and cells were selected randomly from individual embryos for imaging. No blinding to treatment was employed. Unless otherwise noted, all experiments were performed for at least three replicates, each consisting of a minimum three embryos per condition.

### Imaging, FRAP and image analysis

Embryos expressing fluorescent proteins were fixed at stage 26 with 1x MEMFA (0.1 M MOPS, 2 mM EGTA, 1 mM MgSO₄, 3.7% formaldehyde, pH7.4) for 40 min at stage 26, washed with PBS and then imaged. For live images, *Xenopus* embryos were mounted between cover glass and submerged in 1/3x MMR at stage 25–28. Imaging was performed on a Zeiss LSM700 laser scanning confocal microscope using a plan-apochromat 63 × 1.4 NA oil objective lens (Zeiss) or Nikon eclipse Ti confocal microscope with a 63 × 1.4 oil immersion objective. For FRAP experiments, a region of interest (ROI) was defined for full bleach experiments as a 1.75 µm× 1.75 µm box and for half-bleach experiments as a 0.8 µm× 0.4 µm box. ROIs were bleached using 50% laser power of a 488 nm laser and a

0.64 μs pixel dwell time. Fluorescence recovery was recorded at ~0.20 s intervals for up to 300 frames. Bleach correction and normalization was carried out using a custom python script (modified from http://imagej.net/Analyze_FRAP_movies_with_a_Jython_script). Plots were generated using the ggplot2 package in R. 3D projections were generated in Fiji or IMARIS.

## 3D object-based co-localization algorithm

This algorithm and the steps within were primarily implemented using scikit-image (*van der Walt et al., 2014*). First, we segment the cell across all z-stack layers. An Otsu threshold (*Otsu, 1979*) is applied to each layer in the GFP channel. All pixels above threshold in each image are assigned a value of 1, and all background pixels are assigned a value of 0. These thresholded layers are then summed (projected) into a single image. Each disjoint island of positive pixel values represents a projected 3D object, and the sum of pixel values in each island is a proxy of its volume. We identify the object with the largest pixel sum (volume) as the cell, and hence select the thresholded areas in each z layer as being part of the cell if they projected to this largest object. After thus identifying the cell's cross-section in each layer, we apply morphological closing and opening to each image to remove small features caused by noise. We then apply a second round of Otsu thresholding within each thresholded area to identify intracellular vacuoles and organelles that have a significantly lower concentration of GFP. This final segment mask obtained from the GFP channel is used to identify the cell in both channels; areas outside of this mask in either channel are no longer relevant to downstream processing. Second, we isolate cellular areas with a significantly higher than average fluorescence by subtracting the fluorescent background present across the cell. This is performed for both channels separately. We apply a median filter to the cell, effectively subtracting the background fluorescence, and then apply morphological opening and closing to remove noise. One of the resulting areas frequently over-represented using this approach as an area of enriched fluorescence is the cell boundary: quite often, the cell's edges are significantly more fluorescent than the rest of the cell. We attenuate this effect by multiplying areas near the edge by a factor linearly interpolated between 0 and 1 with increasing distance from the edge: that is at the edge values are multiplied by 0, at a distance of 20 pixels from the edge values are multiplied by 0.5, and at a distance of 40 pixels from the edge values are multiplied by 1. Below, we will call the resulting image the *fluorescent foreground*. Third, the Laplacian of Gaussian (LoG) is applied to the fluorescent foreground to identify puncta in both GFP and RFP channels. Fourth, we segment each punctum's body to accurately capture its intensity. LoG identifies the puncta, but it does not accurately capture their size or intensity; we need to identify which pixels are contiguous to each punctum and thus are a part of it. To do this, we apply the watershed algorithm to the fluorescent foreground, with each LoG identified punctum acting as a watershed marker. Fifth, we measure the intensity of each punctum by summing the fluorescent foreground pixels in each watershed basin. Sixth, we collate these 2D puncta from each z-layer into 3D puncta. We do this by simply querying whether puncta watersheds obtained in step four overlap in adjacent z-layers. Seventh, we measure the overlap between 3D puncta in RFP and GFP channels. For a punctum in the RFP channel, the overlap metric is its area overlapping any GFP punctum divided by its total area. Both overlapped and total areas are summed for each punctum across all z-layers in which it is present, thus acting as a volume-based overlap metric. For GFP channel puncta, this is computed analogously. Finally, we create a background overlap rate by randomly shuffling puncta positions in each cell and computing their overlaps using the same metric. Puncta in each channel are randomized independently. Randomization is performed for each punctum one at a time. For each punctum, a random set of coordinates is chosen in 3D space. Coordinates in the image plane are pixel coordinates; z-stack coordinates are the discrete layers. Once a random triplet of coordinates is chosen, the entire 3D punctum is placed at the new position by simply shifting all of its components in each layer by the same amount in the 2D planes, and then shifted a discrete number of layers up or down in the z direction as appropriate. We check whether any part of the punctum lies outside of the cell segmentation found above and whether it overlaps any previously placed punctum. If either is true, we generate new random coordinates again. We continue until all puncta have been placed or a computational limit is reached (in practice, all puncta were placed successfully). For each cell, we generated ten such synthetic replicates to obtain background overlap rates for each cell. Code for this algorithm has been deposited in GitHub (*Boulgakov, 2018*).

## Immunostaining

*Xenopus* embryos were fixed at stage 25 by cold Dent's fixative (80% methanol +20% DMSO) overnight and then were transferred in 100% methanol. Embryos were rehydrated consecutively with TBS (155 mM NaCl, 10 mM Tris-Cl, pH7.4) and then were blocked in 10% FBS, 5% DMSO in TBS. Monoclonal anti-acetylated alpha-tubulin antibody (Sigma-Aldrich, T6793, RRID:AB_477585, 1:500 dilution) and rabbit polyclonal anti-Ruvbl2 (Abcam, ab91462, RRID:AB_2050278, 1:1000 dilution) antibody were used as primary antibodies. Primary antibodies were detected by FITC-goat anti-rabbit antibody (Sigma-Aldrich, F9887, RRID:AB_259816, 1:400 dilution) and AlexaFluor 555-anti-mouse IgG antibody (Invitrogen, A21422, AB_141822, 1:400 dilution).

Human airway cells were fixed and immunostained as previously described using primary and secondary antibodies.(*Pan et al., 2007*; *You et al., 2002*) Primary antibodies and dilution used included rabbit monoclonal LRRC6 (Sigma-Aldrich, HPA028058, RRID:AB_1853337, 1:100 dilution), mouse monoclonal DNAI1(Neuromatb, cat# 73–372, RRID:AB_2315828, 1:2000 dilution) and mouse monoclonal acetylated $\alpha-$tubulin (Sigma-Aldrich, RRID: T7451, clone 6–11-B1, RRID:AB_609894, 1:5000). Primary antibodies were detected using fluorescently labeled, species-specific donkey antibodies (Alexa Fluor, Life Technologies, Grand Island, NY, USA). Nuclei were stained using 4', 6-diamidino-2-phenylindole (Sigma-Aldich).

## Airway epithelial cell culture

Human airway epithelial cells were isolated from de-identified surgical excess of trachea and bronchi removed from lungs donated for transplantation. The use of these cells was exempt from human studies by the Institutional Review Board at Washington University School of Medicine. Tracheobronchial epithelial cells were expanded in culture, seeded on supported membranes (Transwell, Corning Inc., Corning, NY), and differentiated using air-liquid interface conditions as previously described and maintained in culture for up to 10 weeks.

## RT-PCR

To verify the efficiency of *HEATR2* MOs, both L and S MOs were injected into all cells at the 4 cell stage and total RNA was isolated using the TRIZOL reagent (Invitrogen Cat#15596026) at stage 25. cDNA was synthesized using M-MLV Reverse Transcriptase (Invitrogen, Cat# 28025013) and random hexamers. Heatr2 cDNAs were amplified by Taq polymerase (NEB, M0273S) with these primers:

Heatr2.L 25F GCGACTTCCGATGTGACTAA
Heatr2.L 661R CTTCCCACTGCTGTACTGTATAA
Heatr2.S 649F GGCAATGGAAAGTCCGTAGAT
Heatr2.S 1058F CAACAACCCAGTCCGTTACA
Heatr2.L 471F CTTCCCAGAGGTGAAGAAAGAG
Heatr2.L 944R GAAGGACATGGAGCACTGAA

## Acknowledgments

We thank D Dickinson for critical reading and comments on the manuscript, K Drew for technical assistance, and Z Sun for the gift of the Ruvbl2 antibody. The authors acknowledge the Texas Advanced Computing Center (TACC) at The University of Texas at Austin for providing HPC resources that have contributed to the research results reported within this paper. This work was supported by grants from the NIH (R01 HL117164; R21 GM119021, R01 HD085901 to JBW and/or EMM; and DP1 GM106408, R01 DK110520, R35 GM122480 to EMM and NIH HL128370 to SLB), the ATS Foundation/Primary Ciliary Dyskinesia Foundation/Kovler Family Foundation (to AH), NSF (DGE-1610403, to AAB), the Welch foundation (F-1515, to EMM), and a Supplement to Promote Diversity in Health-Related Research from the NICHD (to JBW/RLH).

## Additional information

### Funding

| Funder | Grant reference number | Author |
|---|---|---|
| National Heart, Lung, and Blood Institute | | Ryan L Huizar<br>Chanjae Lee<br>John B Wallingford |
| American Thoracic Society | PCD/Kovler Family foundation | Amjad Horani |
| National Heart, Lung, and Blood Institute | HL128370 | Steven L Brody |

The funders had no role in study design, data collection and interpretation, or the decision to submit the work for publication.

### Author contributions

Ryan L Huizar, Conceptualization, Formal analysis, Validation, Investigation, Visualization, Methodology, Writing—review and editing; Chanjae Lee, Conceptualization, Data curation, Formal analysis, Supervision, Validation, Investigation, Visualization, Methodology, Writing—review and editing; Alexander A Boulgakov, Software, Investigation, Methodology; Amjad Horani, Formal analysis, Funding acquisition, Investigation, Visualization, Methodology, Writing—review and editing; Fan Tu, Formal analysis, Investigation, Methodology; Edward M Marcotte, Software, Supervision, Funding acquisition, Project administration, Writing—review and editing; Steven L Brody, Supervision, Funding acquisition, Investigation, Methodology, Writing—review and editing; John B Wallingford, Conceptualization, Supervision, Funding acquisition, Visualization, Writing—original draft, Project administration, Writing—review and editing

### Author ORCIDs

Ryan L Huizar http://orcid.org/0000-0003-4695-9674
Alexander A Boulgakov http://orcid.org/0000-0002-7446-1120
Amjad Horani http://orcid.org/0000-0002-5352-1948
Edward M Marcotte http://orcid.org/0000-0001-8808-180X
John B Wallingford http://orcid.org/0000-0002-6280-8625

### Ethics

Animal experimentation: Work here was approved by the UT Austin IACUC under protocol numbers: AUP-2015-00160 and AUP-2016-00184.

### Decision letter and Author response

Decision letter https://doi.org/10.7554/eLife.38497.033
Author response https://doi.org/10.7554/eLife.38497.034

## Additional files

### Supplementary files

• Transparent reporting form
DOI: https://doi.org/10.7554/eLife.38497.029

### Data availability

Software has been deposited to GitHub (https://github.com/marcottelab/FociFinder3D; copy archived at https://github.com/elifesciences-publications/FociFinder3D) and image data has been uploaded to Dryad (https://doi.org/10.5061/dryad.j3p00pc).

The following dataset was generated:

| Author(s) | Year | Dataset title | Dataset URL | Database and Identifier |
|---|---|---|---|---|
| Huizar RL, Lee C, Boulgakov AA, Horani A, Tu F, Marcotte EM, Brody SL, Wallingford JB | 2019 | Image data from A liquid-like organelle at the root of motile ciliopathy | https://doi.org/10.5061/dryad.j3p00pc | Dryad , 10.5061/dryad.j3p00pc |

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
