## [Decision Letter]

Thank you for submitting your article "A liquid-like organelle at the root of motile ciliopathy" for consideration by *eLife*. Your article has been reviewed by three peer reviewers, and the evaluation has been overseen by a Reviewing Editor and Anna Akhmanova as the Senior Editor. The following individuals involved in review of your submission have agreed to reveal their identity: Mary E Porter (Reviewer #3).

The reviewers have discussed the reviews with one another and the Reviewing Editor has drafted this decision to help you prepare a revised submission.

The study by Huizar et al. is an elegant demonstration that many of the so-called dynein axonemal assembly factors (DNAAFs) co-localize with axonemal dynein subunits in cytoplasmic foci in multiciliated cells (MCCs) of the human airway (Figure 1—figure supplement 1) and *Xenopus* embryos (Figure 1). This manuscript focuses in particular on studying this phenomenon in the *Xenopus* embryo, which is more experimentally tractable than human airway cells. Using robust statistical measurements for co-localization, they demonstrate that the DNAAFs co-localize with axonemal dynein subunits, whereas other cytoplasmic fusion proteins do not (Figure 1—figure supplements 2, 3, 4). They name these cytoplasmic foci DynAPs or dynein axonemal particles. These authors have described the behavior of the DynAPs as liquid-like phase separated organelles. What controls the formation and dynamics of these organelles is still unknown, and their exact function, beyond concentrating cytoplasmic proteins in a discrete location, is also unknown. However, the demonstration that these foci exist is exciting and opens the door for future study of their formation, behavior, function, and regulation.

While all the reviewers appreciated the technical quality of the paper, and the rigorous statistical analysis, they had some concerns. However, I feel confident that you can answer these concerns and look forward to seeing a revised manuscript.

In particular, please pay attention to point 9 below, concerning the off target effects of morpholinos, which are well documented as a problem that requires extensive controls. There is no western blot or immunostaining shown for *HEATR2* in the MO-treated cells, this control needs to be included.

1) Membrane-RFP stains the axonemal and plasma membrane in *Xenopus* ciliated cells. What are the red foci that are highlighted by the membrane-RFP staining in Video 1? What is membrane-RFP and why should it light up equivalently sized, presumably membrane bound, structures alongside the fluid phase organelles – what do these contain and look like for their content?

2) Tia1 results seem a bit confusing though explained at some length, because we are told it is typically not present in DynAPs but we can see it present in them, in Figure 1—figure supplement 4E, whilst Figure 1—figure supplement 4E versus Figure 1H graph do not seem to correlate together. The immunofluorescence suggests some level of co-localization but we are not clear how much – it could be helpful to draw a line of cut off in Figure 1H, showing where this decision is made regards to whether a protein is considered to co localize or not with the KTU. What about subcellular fractionation or coIP data, to look in more quantified way at these other liquid phase compartment markers?

3) The Results state '200 FP-fusions that report specific localization patterns in *Xenopus* MCCs but do not co-localize with DNAAFs'. This hasn't been done using a co-localization scheme on this scale in Tu et al. (2017), so this should be edited to make clearer this is based upon not having a foci-type localization pattern only, not on co-localization with DNAAF data, if that is the case.

Related to this: "Consistent with this idea, we found that DynAPs were present not only in mature MCCs but also in nascent MCCs undergoing ciliogenesis (Figure 7), and we also observed DynAP in immature human airway MCCs prior to ciliogenesis (Figure 1—figure supplement 1)." It seems that the number of DynAPs in the mature MCC is reduced. Authors should provide quantification of the number DynAP in nascent and mature MCCs to show if there is a difference in numbers of DynAP between cells undergoing ciliogenesis and mature MCCs.

4) Video 2 illustrates a distinct localization for endogenous Ruvbl2 and the tubulin-axonemes, but Ruvbl2 is a multifunctional co-chaperone complex protein presumably working on recruiting HSPs to multiple other events than cilia assembly. As the text mentions later, Ruvbl2 is a component of stress granules for example. How can we account for this very specific staining of the protein and not a more widespread localisation – does this differ according to ciliation status and different cell types?

5) In Figure 3A, what is the 'membrane' label marker exactly? As no other cell lineage-specific markers are shown how do we know the other cells are all goblet cells – does the *Xenopus* epithelium contain only two cell types the MCC and goblet cells?

6) Figure 3C shows the ectopic DynAPs quite well in panel c' but in panel C it is not very clear that motile cilia have been labelled here – please show in higher mag or else add in a specific tubulin staining or similar axonemal marker to make this clear the overlap between cilium and DynAP.

7) Figure 3D is useful on the relationship between cilia versus DynAP number. Can an additional quantification be graphed, to show cilia versus DynAP number in each of these 3 cell types, WT and Mcidas/Foxj1 treated? Mcidas overexpression does not seem to correlate with significantly increased DynAP number, unlike the clearly increased cilia numbers, does it mean there is an upper threshold of some kind limiting their numbers.

8) Figure 7 – an important question needs to be better measured in this superior model system, over 100 cells or similar – to quantify more precisely the number of DynAP vs. cilia length/number.

9) Off target effects for morpholinos are well documented as a problem that requires extensive controls. There is no western blot or immunostaining shown for *HEATR2* in the MO-treated cells, this control needs to be included.

10) Figure 8A needs the accompanying video supplied. Figure 8G – how were DynAPs quantified in this experiment, how were they visualised? It is hard to reconcile how DNAI2 is clearly reduced in the immunofluorescence images of the HEATR2 KD cells, but the western blot does not show reduced levels of protein. How is this explained? Is it to be believed that the levels of DNAI2 in Figure 8d' are the same as those in Figure 8c'?

11) What exactly is reduced mobility of KTU testing, in Figure 8H – how was this tested, what is the experiment? Just make it clearer here.

12) Subsection “DynAPs are distinct from stress granules but share a subset of molecular machinery”, first paragraph: The authors show that G3bp1 co-localizes with Ktu. Did the authors also observe co-localization of G3bp1 with dynein arm components?

---

## [Author Response]

[…] While all the reviewers appreciated the technical quality of the paper, and the rigorous statistical analysis, they had some concerns. However, I feel confident that you can answer these concerns and look forward to seeing a revised manuscript.In particular, please pay attention to point 9 below, concerning the off target effects of morpholinos, which are well documented as a problem that requires extensive controls. There is no western blot or immunostaining shown for HEATR2 in the MO-treated cells, this control needs to be included.1) Membrane-RFP stains the axonemal and plasma membrane in Xenopus ciliated cells. What are the red foci that are highlighted by the membrane-RFP staining in Video 1? What is membrane-RFP and why should it light up equivalently sized, presumably membrane bound, structures alongside the fluid phase organelles – what do these contain and look like for their content?

The memRFP is simply a CAAX domain fused to RFP; it labels the plasma membrane as well as diverse endo-membranes. This has been clarified in the first paragraph of the subsection “DynAPs display hallmarks of biological phase separation”.

2) Tia1 results seem a bit confusing though explained at some length, because we are told it is typically not present in DynAPs but we can see it present in them, in Figure 1—figure supplement 4E, whilst Figure 1—figure supplement 4E versus Figure 1H graph do not seem to correlate together. The immunofluorescence suggests some level of co-localization but we are not clear how much – it could be helpful to draw a line of cut off in Figure 1H, showing where this decision is made regards to whether a protein is considered to co localize or not with the KTU. What about subcellular fractionation or coIP data, to look in more quantified way at these other liquid phase compartment markers?

There is in fact no clear consensus on how to establish significance for co-localization data (see Dunn et al., 2011). Thus, to establish which markers were and were not present in DynAPs, we chose a multiple ANOVA test to compare the Pearson correlation coefficients for all combinations. As shown in Figure 1H, all of the markers we designate as "not present" in DynAPs display significantly lower Pearson correlation values as compared to all markers we designate as "present." Accordingly, as shown in Figure 6B, G3bp1 did *not* differ from the DynAP markers but *did* significantly differ from Tia1, so we designate Tia1 is "not present." This has been clarified.

That said, the reviewer brings up an important point: the new manuscript highlights recent large-scale proteomic analyses of P bodies and stress granules that revealed substantial overlap (Aizer et al., 2008; Feric et al., 2016; Jain et al., 2016; Kedersha et al., 2005) and discusses the fact that while DynAPs appear distinct from stress granules they do share some machinery (subsection “DynAPs are distinct from stress granules but share a subset of molecular machinery”).

Finally, fractionation is an excellent idea but will first demand that we perfect sorting of MCCs from *Xenopus* and then perfecting the fractionation and then detecting the proteins (for which antibodies do not cross react in *Xenopus*). Also, co-IP may or may not reveal interactions, as many proteins in liquid like organelles do not make stable associations; this experiment may require BioID or Apex. We hope the reviewer agrees that this would be beyond the scope the initial characterization in this paper.

3) The Results state '200 FP-fusions that report specific localization patterns in Xenopus MCCs but do not co-localize with DNAAFs'. This hasn't been done using a co-localization scheme on this scale in Tu et al. (2017), so this should be edited to make clearer this is based upon not having a foci-type localization pattern only, not on co-localization with DNAAF data, if that is the case.

This is a good point, and we regret overstating our case. We have made the change.

Related to this: "Consistent with this idea, we found that DynAPs were present not only in mature MCCs but also in nascent MCCs undergoing ciliogenesis (Figure 7), and we also observed DynAP in immature human airway MCCs prior to ciliogenesis (Figure 1—figure supplement 1)." It seems that the number of DynAPs in the mature MCC is reduced. Authors should provide quantification of the number DynAP in nascent and mature MCCs to show if there is a difference in numbers of DynAP between cells undergoing ciliogenesis and mature MCCs.

We have noted no difference in DynAP numbers between stages, so we replaced this image with one that is more representative. We feel that careful quantification of the time course of DynAPs is beyond the scope of the present work, and we hope the reviewers agree.

4) Video 2 illustrates a distinct localization for endogenous Ruvbl2 and the tubulin-axonemes, but Ruvbl2 is a multifunctional co-chaperone complex protein presumably working on recruiting HSPs to multiple other events than cilia assembly. As the text mentions later, Ruvbl2 is a component of stress granules for example. How can we account for this very specific staining of the protein and not a more widespread localisation – does this differ according to ciliation status and different cell types?

The reviewer is correct that Ruvbl2 is a broadly acting chaperone, and we regret that the initial submission was not explicit about this point and did not place it in the proper context:

A close look at Figure 1—figure supplement 5 reveals that Ruvbl2 is present in large foci that are specific to MCCs (DynAPs), but Ruvbl2 is also present in much smaller foci in both MCCs and the neighboring goblet cells. This result is in fact consistent with previous reports of Ruvbl2 as a foci-forming protein (Rizzolo et al., 2017). We now discuss that previous work and explicitly describe this result in the revision (subsection “DynAPs concentrate core Hsp70/90 chaperones and specific co-chaperones”). We also note that this result is less apparent in the projections in Video 2, and we have added a note to the legend of Video 2 directing the reader to Figure 1—figure supplement 5.

We also hasten to add that genetic data from mouse and zebrafish reveal a surprisingly specific role for Ruvbl2 in dynein assembly: Xiaoxia Sun's lab at Yale has shown that MCCs that are mutant for this gene develop largely normally and make multiple cilia, but that these cilia fail to beat and lack dynein arms.

Ultimately, while surprising, both their functional genetic data and our localization data argue for a surprisingly specific role for Ruvbl2 in MCCs.

5) In Figure 3A, what is the 'membrane' label marker exactly? As no other cell lineage-specific markers are shown how do we know the other cells are all goblet cells – does the Xenopus epithelium contain only two cell types the MCC and goblet cells?

We regret that we failed to explain this assay adequately. There are, in fact, four cell types in the epidermis: MCCs, goblet cells, ionocytes and small secretory cells. The latter two cell types are easily apparent by the their far smaller apical surface areas. More importantly, only the MCCs are ciliated. We now describe this assay more clearly, and as per comment 6, we have repeated this experiment with a specific marker of motile cilia and replaced the figure. These changes are in the subsection “DynAPs are MCC-specific organelles that assemble under the control of the motile ciliogenic transcriptional circuitry”.

6) Figure 3C shows the ectopic DynAPs quite well in panel c' but in panel C it is not very clear that motile cilia have been labelled here – please show in higher mag or else add in a specific tubulin staining or similar axonemal marker to make this clear the overlap between cilium and DynAP.

We have repeated this experiment with a specific marker of motile cilia.

7) Figure 3D is useful on the relationship between cilia versus DynAP number. Can an additional quantification be graphed, to show cilia versus DynAP number in each of these 3 cell types, WT and Mcidas/Foxj1 treated? Mcidas overexpression does not seem to correlate with significantly increased DynAP number, unlike the clearly increased cilia numbers, does it mean there is an upper threshold of some kind limiting their numbers.

While we agree that the issue of DynAP numbers will be important, accurately counting individual cilia from confocal images is challenging, and we feel this question is beyond the scope of this initial characterization of the DynAPs. We hope that the reviewers will agree.

8) Figure 7 – an important question needs to be better measured in this superior model system, over 100 cells or similar – to quantify more precisely the number of DynAP vs. cilia length/number.

As noted above, we feel that the issue of DynAP numbers would be an excellent topic for a forthcoming paper but is beyond the scope of the current paper, which focuses on describing this novel organelle.

9) Off target effects for morpholinos are well documented as a problem that requires extensive controls. There is no western blot or immunostaining shown for HEATR2 in the MO-treated cells, this control needs to be included.

We fully agree that MOs require extensive controls, specifically to rule out off-target effects. In light of this serious concern, we have added new data to address both efficacy and off-target issues with our MOs.

First, we respectfully point out that the suggested control of a western blot –while useful – does not address off-target effects, but rather demonstrates efficacy of the reagent to disrupt the target. However, we were unable to use western blotting because existing Heatr2 antibodies did not cross-react in *Xenopus*. As a proxy, we performed RT-PCR to validate the efficacy of our MO (Figure 8—figure supplement 1), and showed that injection of the MOs led to both disruption of splicing *and* a near total loss of the Heatr2 mRNA (a common effect of splice-blocking MOs due to nonsense mediated mRNA decay).

To address the off-target issue directly, we also repeated our experiment using a second, non-overlapping set of MOs that target different splice sites in the L and S alleles of *heatr2*. Like the first MO set, the second MOs disrupted splicing and reduced mRNA levels of *heatr2* and also elicited reductions in DynAP numbers as well as disrupting deployment of dyneins to axonemes.

Finally, we note that both MOs precisely recapitulate the phenotype observed in humans with genetic mutations in *HEATR2* (e.g. normal length cilia lacking axonemal dyneins).

We now feel very confident in these results, and we hope the reviewers do as well.

10) Figure 8A needs the accompanying video supplied. Figure 8G – how were DynAPs quantified in this experiment, how were they visualised? It is hard to reconcile how DNAI2 is clearly reduced in the immunofluorescence images of the HEATR2 KD cells, but the western blot does not show reduced levels of protein. How is this explained? Is it to be believed that the levels of DNAI2 in Figure 8d' are the same as those in Figure 8c'?

We regret not being clear. Figure 8c' and 8d' show only the apical slices of the Z-stack to focus attention on the cilia, which almost entirely lack dyneins in the Heatr2 morphant, while Figure 8E and 8F show slices of similar cells but focusing on the cytoplasm, where dynein is present in both controls and knockdowns. DynAPs were quantified by counting foci in the cytoplasm. We have made this clearer in the revised text.

11) What exactly is reduced mobility of KTU testing, in Figure 8H – how was this tested, what is the experiment? Just make it clearer here.

This is the mobile fraction in FRAP assays; this has been clarified.